# Analytical Research on the Impact Test of Light Steel Keel and Lightweight Concrete of Composite Wall

**Jianyu Yang, Jiaming Zou and Weijun Yang \***

School of Civil Engineering, Changsha University of Science and Technology, Changsha 410014, China; jianyuy@csust.edu.cn (J.Y.); 21002020044@stu.csust.edu.cn (J.Z.)
* Correspondence: yyyaozhijian@163.com

**Abstract:** In order to study the impact resistance of light steel keel and lightweight concrete of composite walls (LSKLCW) under low-velocity impact, four composite wall specimens were designed to conduct dynamic simulation impact tests, and the failure mode, time-history curves of strain and displacement were analyzed and studied using test equipment and a loading system. The results show that the failure characteristics of the composite wall sample were elastic–plastic. Moreover, the vertical displacement and strain at the most unfavorable collision point were linearly related to the impingement height. Furthermore, the capacity of the composite wall (such as crack resistance, elastic–plastic deformation and energy dissipation) was affected by the concrete strength and the arrangement of the light steel netting. In addition, the impact resistance of the wall was significantly improved when the concrete strength was enhanced and the light steel netting was installed. Lastly, the test results were fitted and verified through the impact force calculation model of the composite wall, and then the accuracy of the test model was analyzed. The certain experimental basis and theoretical analysis basis for the impact resistance research of the composite wall can be provided by these research results.

**Keywords:** composite wall; impact; experimental study; failure modes; time-history analysis

## 1. Introduction

Nowadays, the construction industry is developing and innovating in the direction of green energy saving, environmental protection and safety. LSKLCW have been widely used in construction in recent years because of their better fire resistance, heat and sound insulation and other performance aspects. Scholars have carried out preliminary research and exploration on multiple types and different combinations of composite walls. Research has shown that composite walls have good deformation or energy dissipation capabilities and other excellent performance characteristics [1–5]. In addition, the research field of the impact resistance of composite walls is gradually attracting more attention. Scholars at home and abroad have carried out a series of work in the field of impact [6–11]. However, there are few studies on the impact resistance of composite walls under low-speed impact and the variation law of impact force.

During the engineering application process, the composite wall will experience various impact effects, such as collision in the process of installation and transportation, impact, explosion and external impact. Among them, the building safety of composite walls will be affected by low-speed impacts. Therefore, studying the effect of impact on its performance and the law of impact force change has practical guiding significance for the impact research of composite walls. In this paper, four composite wall specimens were produced to study the impact resistance of LSKLCW, and the dynamic measurement method [12] was adopted to collect the date of the impact test. In addition, the failure form, strain time-history curve and displacement time-history curve of the composite wall were collected and analyzed after the impact test. Moreover, based on the law of conservation of kinetic energy and the

momentum theorem, a numerical model was established and the accuracy of the model was verified by the test results. Finally, the result provides a certain experimental basis and theoretical analysis basis for the impact resistance research of composite walls.

## 2. Test Overview

### 2.1. Specimen Design and Production

In order to reasonably simulate the situation of the composite wall under the low-speed impact in the actual scenario and design a comparison test group to study the response of the components inside the composite wall under low-speed impact, four composite wall specimens were designed and manufactured, which were numbered W1, W2, W3 and W4. The design dimensions of the test piece are shown in Figure 1. Three cold-formed thin-walled light steels were arranged inside each test piece as vertical force-bearing skeletons, and the skeleton spacing was set to 300 mm. Moreover, after setting the comparative test parameters, the specific parameters of each wall specimen were as shown in Table 1. The solid model of the specimen after pouring and curing is shown in Figure 2.

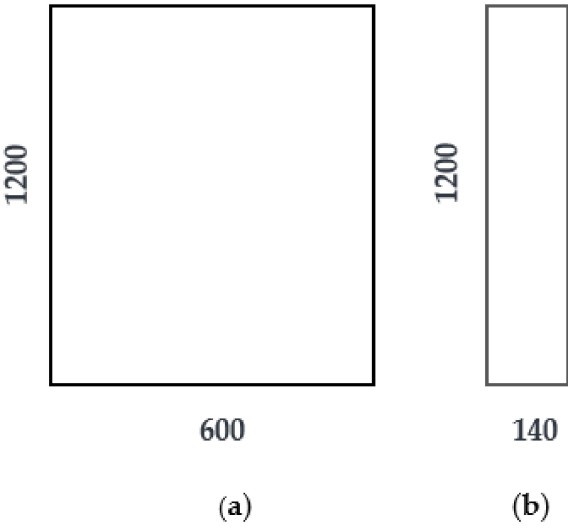

**Figure 1.** Size design drawing of LSKLCW. (**a**) Front size; (**b**) Cross-section size.

**Table 1.** Parameters of LSKLCW test.

| Serial Number | Specimen Number | Section Size, mm | Wall Height, mm | Whether to Set Up Light Steel Mesh | Lightweight Concrete, MPa |
|---|---|---|---|---|---|
| 1 | W1 | 140 × 600 | 1200 | Yes | No setting |
| 2 | W2 | 140 × 600 | 1200 | Yes | 8 |
| 3 | W3 | 140 × 600 | 1200 | Yes | 10 |
| 4 | W4 | 140 × 600 | 1200 | No | 8 |

In Table 1, the selected model type of steel is S350GD, and the yield strength is 435 MPa. Then, the elastic modulus of the steel is $2.06 \times 10^5$ N/mm$^2$, the density is around 7.85 g/cm$^3$, and the coefficient of linear thermal expansion is $1.2 \times 10^{-5}$ °C.

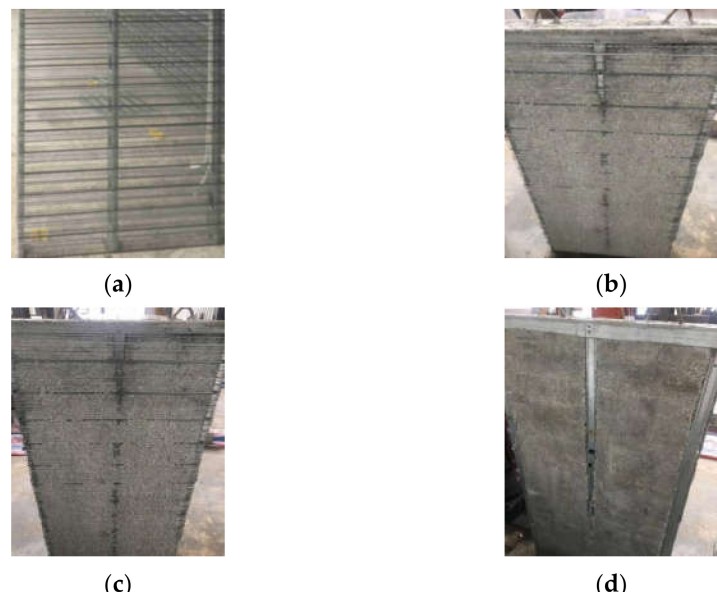

(a)　　　　　　　　　　　　　　　　　　(b)

(c)　　　　　　　　　　　　　　　　　　(d)

**Figure 2.** Solid model diagram of test piece. (**a**) W1; (**b**) W2; (**c**) W3; (**d**) W4.

## 2.2. Test Equipment and Loading System

We selected a DH3822 portable dynamic signal test and analysis system, LVDT digital acquisition recorder, dynamic displacement meter and other dynamic measurement as test equipment. In addition, this test system can quickly feedback the real-time changes in the measurement point strain in the test, and these instruments are used frequently in tests because of their intelligent and convenient mechanism. Furthermore, partial parameters and connection methods of the DH3822 portable dynamic signal test and analysis system are shown in Figure 3. Moreover, the LVDT digital acquisition recorder, including the different inductance-type transducer and Lab-VIEW analysis software, and partial parameters of LVDT are shown in Figure 4.

| Parameter Name | Specification |
|---|---|
| Number of Channel | 16 |
| The rate of Continuous Sampling | WiFi、Ethernet communication mode , Max: 1kHz/one channel |
| The Mode of Bridge Circuit | Full Bridge ; Half Bridge ; Quarter Bridge of Three-Wire Measurement |
| The Voltage of Supply bridge | 2V DC、5V DC、10V DC |
| Strain Range | ±50000με , Min : 0.5με |
| Voltage Range | ±5000mV、±500mV , Minimum Resolution : 5μV |
| Power supply | Interchange 220V |

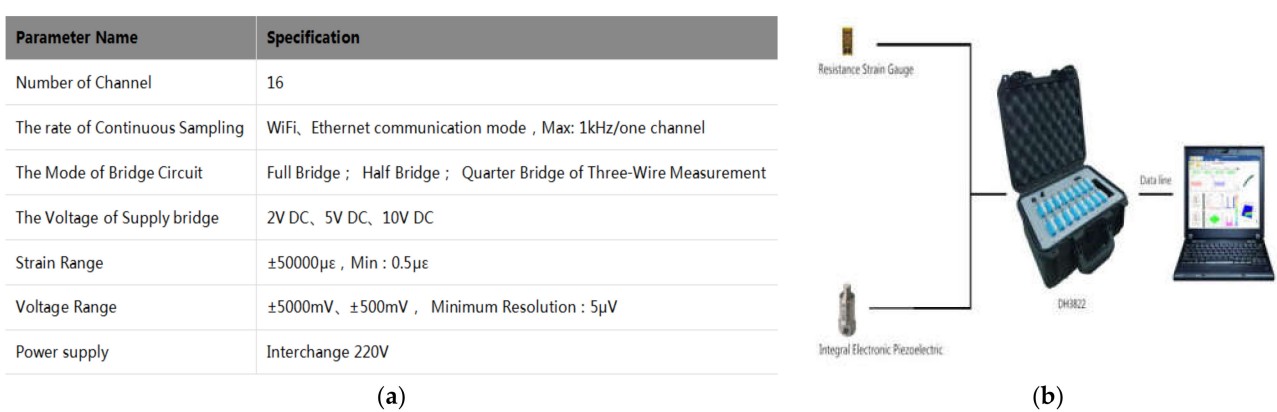

(a)　　　　　　　　　　　　　　　　　　(b)

**Figure 3.** Partial parameters and connection methods of DH3822. (**a**) Parameter; (**b**) Connection method.

In the test, the 30 kg standard sandbag was selected as the collision object and the impact height was chosen as a parameter variable; the loading test was carried out in two groups of different impact heights. In addition, the test process was recorded by the experimental instrument, and measurements of each set of data were separated by 1 h.

| Model | TW01 |
|---|---|
| Capacity | 0-250mm |
| Rated Output | 100-1000 ( mv/v/mm ) |
| Precision | 0.05%F.S 0.1%F.S  0.2%F.S |
| Dynamic frequency | 0-200HZ |
| Temperature effect onoutput | ≤0.02%F.S/°C |
| Temperature effect on zero | ≤0.02%F.S/°C |
| Safe Temperature Range | -20°C+70°C |

**Figure 4.** Partial parameters of LVDT.

### 2.3. Measuring Method and Measuring Point Layout

Furthermore, the low-speed collision process in actual engineering problems can be simulated by the free-release form, which adopts the falling sandbag [13,14]. The arrangement of strain measuring points on the contact surface of the test piece is shown in Figure 5. Moreover, the vertical displacement at the most unfavorable position in the center of the wall specimen under impact was measured by the dynamic displacement meter. They were respectively arranged on the back side of the impact surface of the composite wall. In the vertical displacement test, these measuring points were set on the center and circumference of each test piece. In addition, the layout of the vertical displacement measuring points is shown in Figure 6. Furthermore, all measuring points were arranged after each piece was installed, with the panel and the layout of the test shown in Figure 7.

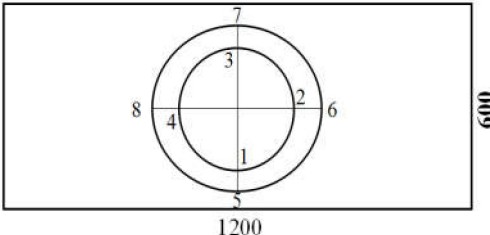

**Figure 5.** Layout drawing of strain measuring point of specimen.

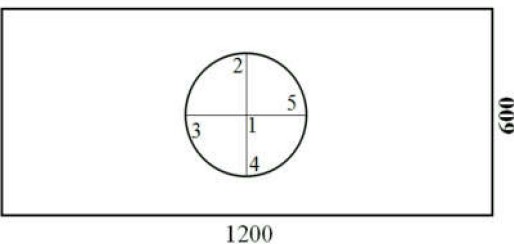

**Figure 6.** Displacement measuring point arrangement on the back side of the specimen.

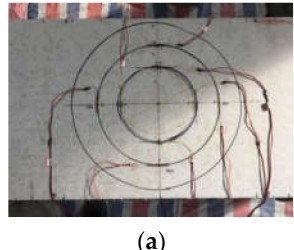

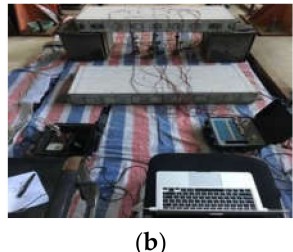

(**a**)

(**b**)

**Figure 7.** Test layout. (**a**) Wire layout; (**b**) Device connection.

In Figure 5, a circle with a diameter of 250 mm and another with a diameter of 350 mm are shown in the center of the test piece, and the strain measuring points were installed on the circumference of each circle. In addition, points 1 to 8 were the strain measuring point positions on the contact surface, where the length and width are given in millimeters.

In Figure 6, points 1 to 5 are the vertical displacement measuring point positions. Among them, point 1 was set at the center of the circle as the most unfavorable point, and points 2 to 4 were set on a circle with a diameter of 250 mm. Length and width are in millimeters.

In Figure 7, the wire layout of one of the specimens as a demonstration is shown in Figure 7a. The device connection of the test is shown in Figure 7b.

## 3. Low-Speed Impact Test Process

The test process of W1 under an impact height of one meter is shown in Figure 6. Before the start of the test, a tape measure was used to focus the center point of collision between the suspended sandbag and the composite wall, as shown in Figure 8a. In addition, the falling process of the sandbag showed that the decreasing rate of the sandbags in unit time increased, as shown in Figure 8b,c. Moreover, a small deformation occurred on the impact area and the compression deformation of the sandbag under the cushioning extrusion of the impact surface could be observed by the naked eye, as shown in Figure 8d.

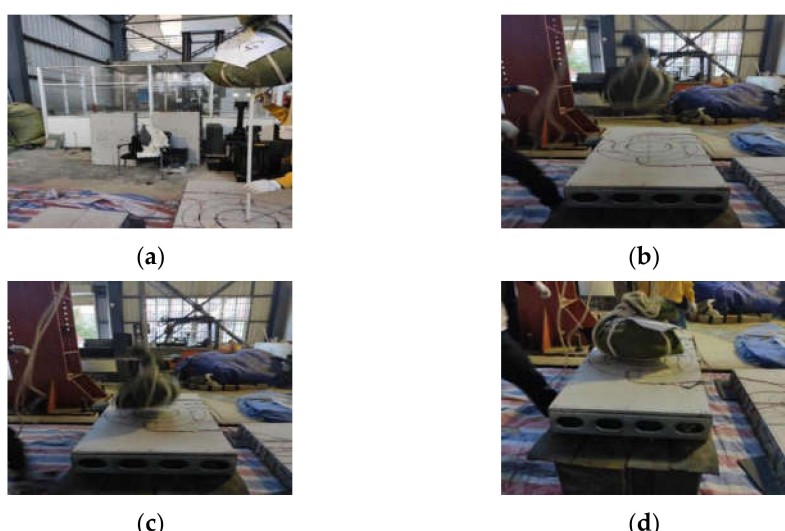

(a)

(b)

(c)

(d)

**Figure 8.** Test process diagram of W1. (**a**) Cross-center correction; (**b**) Falling process; (**c**) Moments before fall contact; (**d**) Collision contact.

In order to reduce the influence of the compressional deformation and make the impact force generated by the falling object more evenly and effectively act on the collision surface, the sandbag was tied up before the impact test of W2, W3 and W4, and then the progress of the W1 test was repeated, as shown in Figure 9. In addition, the failure from of W2, W3, W4 at different impact heights was similar. Since the concrete slurry was poured into these wall specimens, the impact test results showed that the impact resistance of the W2, W3 and W4 wall specimens was significantly improved. In addition, after the impact test, there was no obvious deformation or visible cracks or depressions on the collision surfaces of these test pieces, and these specimens showed better overall stability.

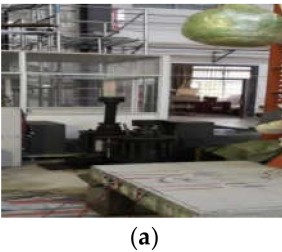 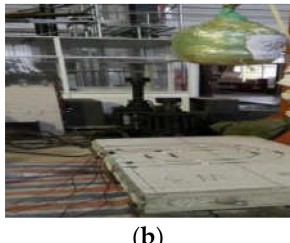 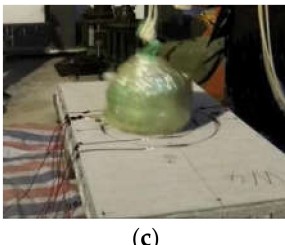
(**a**)　　　　　　　　(**b**)　　　　　　　　(**c**)

**Figure 9.** Test process diagram of W2, W3 and W4. (**a**) W2; (**b**) W3; (**c**) W4.

Furthermore, the deformation trend of the specimens under low-speed impact was similar. As the result, the W1 specimen was taken as a typical failure morphology for analysis. The failure form of W1 without grouting concrete inside was more serious than that of W2, W3, W4. Under the impact of the sandbag, the panel sank and cracked, and the panel was obviously sinking at the most unfavorable collision point within the contact range. In addition, the cracks in the impact area of the W1 specimen were densely and widely distributed, spreading out from the center to the surrounding area, as shown in Figure 10.

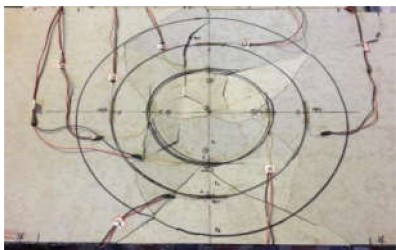

**Figure 10.** Destruction morphology diagram of W1 specimen.

Then, the damage patterns of the W2, W3 and W4 specimens were found to be basically similar, and the damage degree of the wall under the two sets of different impact height tests was not significant. After the collision, the panel had no obvious depression or deformation, and no cracks appeared on the collision surface. Moreover, after these test pieces were subjected to impact, the strain gauges on each measuring point could work normally, as shown in Figure 11. Figure 11 shows that the concrete in the composite wall can well absorb most of the energy generated by the impact, and these specimens only produce small deformations under the action of low-speed impact.

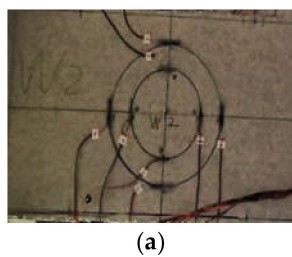 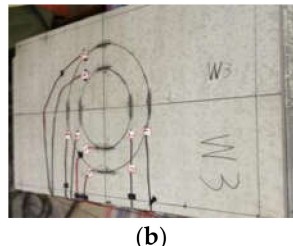 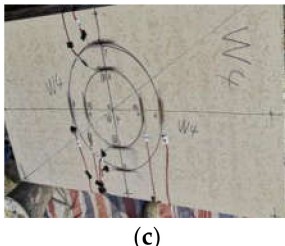
(**a**)　　　　　　　　(**b**)　　　　　　　　(**c**)

**Figure 11.** Damage diagrams of W2, W3 and W4 specimens after being impacted. (**a**) W2; (**b**) W3; (**c**) W4.

## 4. Test Results and Analysis of Time-History Curve

### 4.1. Strain Time-History Curve

According to the data from the test, the strain versus time curve of each measuring point of each test piece under different impact heights was obtained through the data acquisition system and software analysis. The W1 specimen was used as a typical specimen

for experimental analysis under the impact of an impact height of 1 m. The strain time-history curves of each measurement point of the W1 to W4 specimens are shown in Figure 12.

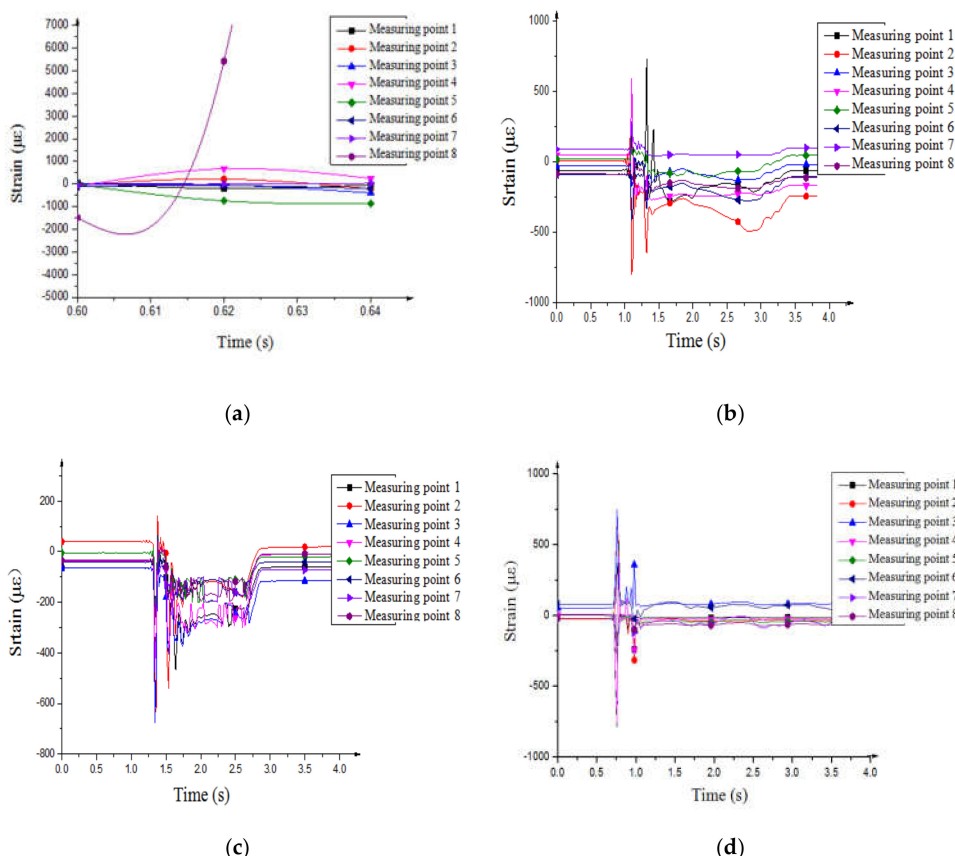

**Figure 12.** Strain time-history curves of W1 to W4. (**a**) W1; (**b**) W2; (**c**) W3; (**d**) W4.

As shown in Figure 12, the changes in strain of each measurement at any moment were recorded by the instrument. In order to distinguish and describe the strain curves for each measured point, the curves were processed through different colors and different shapes of icons.

In Figure 12a, the record starts from the effective contact time of 0.6 s. The damaged W1 specimen was analyzed by the strain time0history curve of other specimens and failure forms. Figure 12 indicates that falling objects collide with the contact surface of the wall within a short contact time in the range of 0.61 s to 0.62 s and the strain of the measuring point increases rapidly. However, the curve of measuring point 8 showed an upward phenomenon and did not appear in a descending stage. The strain gauge was damaged due to the crack development of measuring point 8, which caused the strain change to exceed the measuring range of the strain gauge. The result was basically in line with the test. Furthermore, the strain change from measuring point 1 to measuring point 7 was similar to measuring point 8, but the change was relatively small. Analyzing the failure morphology of W1, the cracking failure from measuring point 1 to measuring point 7 developed rapidly; the panel cracking released most of the energy, and the crack development did not pass through the measurement area of the strain gauge. The results were basically consistent with the test.

Moreover, as shown in Figure 12, the trend of tensile tension and compression of these strain curves was different as these strain gauges were glued to the specimens in a horizontal and vertical manner. In order to obtain the maximal response variable, we only

explored the increment in the strain gauge of each measuring point. The further cases of strain change were analyzed as follows.

Comparing and analyzing the strain time-history curves of W2, W3 and W4 specimens, the time-varying law was similar. In the test, the W4 specimen was taken as a typical specimen for test analysis and description. The strain time-history curves of W4 at the impact height of 1 m and the impact height of 1.5 m are shown in Figure 13.

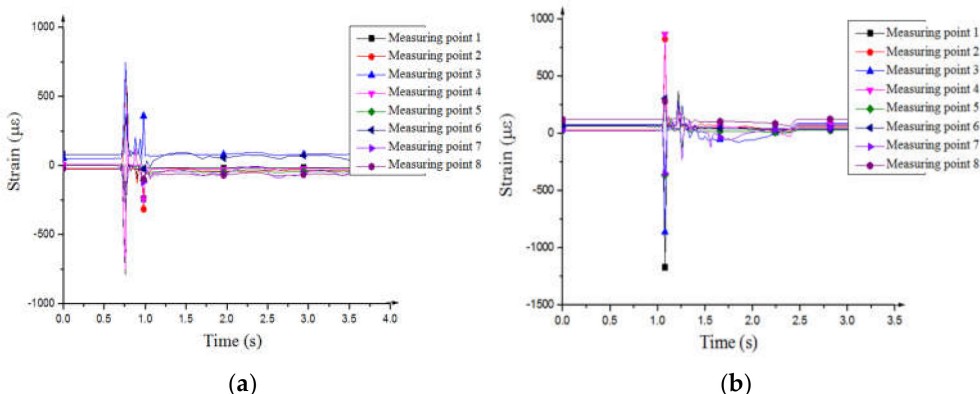

(**a**)                                               (**b**)

**Figure 13.** Strain time-history curve of W4 specimen; (**a**) Impact at a height of 1 m; (**b**) Impact at a height of 1.5 m.

Figure 13 illustrates that the impact height of the sandbag was proportional to the degree of strain change. The strain at the measuring point changed sharply until reaching the top, and then it decreased rapidly. After the wall was impacted, the elastic deformation was recovered slowly, the strain decay rate gradually decreased, and the change in the strain value gradually became stabilized after a short period of aging. The strain value after the test of each measuring point was different from the initial strain, which indicates that a certain small residual deformation occurred after the composite wall was impacted.

Through comparative analysis of the results of the strain time-history curves of each specimen, the strain change degree of W4 was found to be higher than that of W2 and W3. The analysis showed that the influencing factors were mainly related to the light steel formwork and the strength of the poured concrete. Comparing W2 and W4, the W2 test piece was compared with the W4 test piece by adding a light steel mesh, which led to an increase in the impact resistance of the composite wall test piece. Therefore, the strain variation of the strain measuring point on the panel of W2 was smaller than that of W4. Comparing W2 and W3, the concrete strength of W3 was higher than that of W2, which resulted in W3 having better impact resistance. The deformation ability and measured strain change degree of W3 were lower than those of W2.

*4.2. Vertical Displacement Time-History Curve*

According to the analysis of the test results, the vertical displacement time-history curves of the W2, W3 and W4 specimens had a certain similarity, so the W4 specimen was taken as a typical result for analysis. The displacement time-history curves of W4 at the impact height of 1 m and the impact height of 1.5 m are shown in Figure 14.

It can be seen from Figure 14 that the vertical displacement of the measuring point of W4 under the two different impact heights changed within a short recording time from 0.1 s to 0.2 s. Since measuring point 1 was located at the most unfavorable force position in the center, and measuring points 2, 3, 4 and 5 were arranged on the circumference of a circular contact area with a diameter of 250 mm, the vertical displacement of measuring point 1 was greater than that of other measuring points.

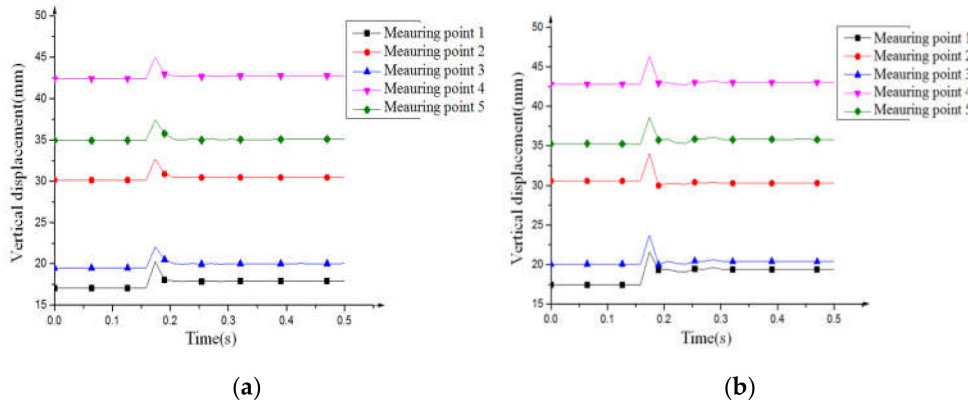

**Figure 14.** The displacement time-history curve of W4 specimen. (**a**) Impact at a height of 1 m; (**b**) Impact at a height of 1.5 m.

Furthermore, the vertical displacement of measuring points 1–5 was measured. It seems that the maximum displacement values were measured at point 1 in theory when the center of gravity of the sandbag coincided with the center of the specimen. This is because measuring point 1 was under the most unfavorable position, and the data of point 1 in Figure 14 confirm this result. In addition, the difference in the displacement value between point 1 and the other points was not large because the power of the low-speed impact was small. Moreover, the displacement values at points 2–5 should be the same in theory. However, there was a deviation in the results due to the effect of a test error and the effect of the dimensions of the specimen. This situation was normal and reasonable.

Moreover, the vertical displacement value recorded after 0.2 s was slightly higher than the vertical displacement value recorded at the initial moment, which was due to the elastic deformation performance of the wall. The recorded results show that there was a dynamic process of slow recovery of deformation after the composite wall was subjected to a low-speed collision. In addition, the vertical displacement change of each measuring point under the impact height of 1.5 m was greater than that under the impact height of 1 m.

By comparing the test results of W2, W3 and W4, it can be seen that the measured maximum vertical displacement of the W3 specimen was smaller than that of W2. The analysis results indicate that the concrete strength of the W3 specimen was higher than that of W2, which led to an increase in the overall strength of the composite wall and enhanced the impact resistance of the W3 specimen to a certain extent.

It can be seen from the vertical displacement curve that the vertical displacement change of the W4 specimen was higher than that of W2. The analysis result shows that W4 was not equipped with light steel mesh, which resulted in the impact resistance of W4 being weaker than that of W2. In addition, the vertical displacement trend of the W2 and W3 specimens under different impact heights was similar to that of W4.

*4.3. Peak Strain at the Center of Collision*

Combined with the test data, the peak strains of the collision centers of the W2, W3 and W4 specimens are shown in Table 2. It can be seen that the peak strain of the collision center point of the specimen in the impact test state was arranged from small to large as W3, W2 and W4.

**Table 2.** Results of peak strain at the center of collision.

| Specimen Number | Impact Height, m | Peak Strain at the Center of Collision, $\mu\varepsilon$ |
|:---:|:---:|:---:|
| W2 | 1 | 864.604 |
| W3 | 1 | 688.905 |
| W4 | 1 | 955.211 |
| W2 | 1.5 | 1084.582 |
| W3 | 1.5 | 886.314 |
| W4 | 1.5 | 1190.876 |

*4.4. Conversion Measurement Result of Maximum Impact Force Value*

The maximum impact test results of W2, W3 and W4 specimens were obtained from the test data, which are shown in Table 3. The W1 specimen was used as a failure model for comparative testing. Therefore, it was not included in the results.

**Table 3.** Results of maximum impact force test.

| Specimen Number | Impact Height, m | Maximum Impact Value, kN |
|:---:|:---:|:---:|
| W2 | 1 | 5.364 |
| W3 | 1 | 4.639 |
| W4 | 1 | 6.872 |
| W2 | 1.5 | 6.704 |
| W3 | 1.5 | 5.982 |
| W4 | 1.5 | 8.797 |

It can be seen from Table 3 that the maximum impact force value was proportional to the impact height, and the calculated values of maximum impact force were arranged in descending order as W4, W2 and W3. In addition, the analysis result shows that the light steel mesh can improve the impact resistance of the composite wall to a certain extent. Moreover, the deformability of the composite wall would be affected by the strength of concrete, and the impact resistance of the wall can be enhanced when the strength of the concrete is increased.

## 5. Validation of Numerical Model of Test Impact Force

The analysis of the test process was based on the law of conservation of kinetic energy and the momentum theorem. In addition, the elastic–plastic deformation state of the composite wall under impact would be considered. Simplified assumptions were made when the experimental model based on the Hertz elastic collision model and Thornton elastic-plastic theory were analyzed [15,16]. It is assumed that the collision between the sandbag and the composite wall satisfies the Hertz contact theory. The assumptions are as follows:

1. The contact surface of the wall panel was regarded as a semi-infinite space.
2. The wall was homogeneous, continuous and isotropic.
3. The sandbag did not rotate.
4. The falling objects of the sandbag were homogeneous and regular spheres.
5. The direction of the sandbag falling speed was perpendicular to the wall panel.

Before the plastic deformation zone was generated on the collision surface of the wall, the condition of elastic contact was satisfied. In the elastic range, the relationship between the elastic contact pressure and the compression of normal deformation can be described by Equation (1), which is as follows:

$$P_e = \frac{4}{3}ER^{\frac{1}{2}}\delta^{\frac{3}{2}} \tag{1}$$

where $P_e$ (kPa) is the pressure of elastic contact; E (N/m$^2$) is the equivalent elastic modulus, $E = \left[\frac{1-\nu_1^2}{E_1} + \frac{1-\nu_2^2}{E_2}\right]^{-1}$; $\nu_1$ is the Poisson's ratio of the sandbag; $\nu_2$ is the Poisson's ratio of the composite wall; $E_1$ (N/m$^2$) is the elastic modulus of the sandbag; $E_2$ (N/m$^2$) is the elastic modulus of the composite wall; R (m) is the equivalent radius, $R = \left(\frac{1}{R_1} + \frac{1}{R_2}\right)^{-1}$; $R_1$ (m) is the radius of the sandbag; $R_2$ (m) is the radius of the wall panel and treated as infinite, $R_2 \rightarrow +\infty$; $\delta$ (m) is the compression of normal deformation.

Based on the Hertz elastic collision model, the radius of the contact surface between two colliding objects r* (m) was assumed. The relationship between the elastic contact pressure, the radius of the contact surface, the amount of deformation and the contact pressure stress during a collision can be described by Equation (2). In addition, the relationship of the radius of the contact surface between the sandbag and the composite wall, the equivalent radius and the compression of normal deformation can be described by Equation (3). These equations are as follows:

$$P(r) = \frac{3P_e}{2\pi r^{*2}}\left[1 - \left(\frac{r}{r^*}\right)^2\right]^{\frac{1}{2}} \tag{2}$$

$$r^* = (R\delta)^{\frac{1}{2}} \tag{3}$$

where P(r) (MPa) is the compressive stress of the contact compression; r* (m) is the radius of the contact surface between the sandbag and the composite wall; r (m) is the variable of the radius of the contact, in the range of (0,r*).

We adopted Thornton's elastic–plastic basic theory as the premise. The yield strength of the contact object was taken as the critical point during collision. The plastic deformation area would be formed on the collision surface when the yield strength of the collision object is less than the limit value of the contact compressive stress. Considering the elastic–plastic nature of the contact material and the material hardening under the dynamic loading conditions (the process of the elastic–plastic change of the material), in order to simplify the calculation process, the elastic–plastic property of the material was simplified to satisfy the hypothetical condition. Furthermore, the relationship between the stress of the yield pressure and the radius of the contact surface at the initial yield time can be established by combining Equations (1)–(3), as shown in Equation (4).

$$p_y = \frac{2Er_y}{\pi R} \tag{4}$$

where $p_y$ (MPa) is the stress of the yield pressure in the plastic area of the collision contact surface; $r_y$ (m) is the radius of the contact surface at the initial yield time.

According to the Hertz contact theory, it is assumed that the falling sandbag satisfies the Von Mises yield criterion [17]. The calculation formula for the stress of the yield pressure in the plastic area of the collision contact surface is as shown in Equation (5).

$$p_y = C_v Y \tag{5}$$

where $C_v$ is the contact coefficient, $C_v = 1.234 + 1.256\nu$; $\nu$ is the Poisson ratio of the contact material; Y (MPa) is the yield strength of the contact material.

Equations (2)–(5) were solved simultaneously to obtain the initial yield compression of the normal deformation $\delta_y$ (m). Substituting the result into Equation (1), we can obtain the initial yield compressive stress of the normal direction $P_y$ (kPa). The contact material can be assumed as an ideal elastic–plastic material by the Thornton theory [18]. In addition, the relationship between the compressive stress of the elastic–plastic contact compression at the normal deformation and the normal compressive deformation was defined, as shown in Equation (6):

$$P_{ep} = P_y + 2\pi R p_y(\delta - \delta_y) \tag{6}$$

where $P_{ep}$ (kPa) is the compressive stress of the elastic–plastic contact compression at the normal deformation; $P_y$ (kPa) is the initial yield compressive stress of the normal direction, $P_y = \frac{4}{3}ER^{\frac{1}{2}}\delta_y^{\frac{3}{2}}$; $\delta_y$ (m) is the initial yield compression of the normal deformation.

Through the analysis of energy conservation, it can be seen that the kinetic energy generated by the falling sandbag was mainly composed of the local elastic deformation energy, the elastic–plastic deformation energy of the contact area and the overall bending deformation energy of the composite wall [19]. Integrating the conservation of momentum into the equation, the maximum compression of the normal deformation under consideration of elastic–plastic deformation can be obtained, as shown in Equation (7):

$$\frac{1}{2}mv_1^2 + \frac{1}{2}mv_2^2 = \int_0^{\delta_y} P_e(\delta)d\delta + \int_{\delta_y}^{\delta_{max}} P_{ep}(\delta)d\delta + \frac{24P_{ep}^2\left(1 - v_2^2\right)}{\pi^4 E_2 h^3 ab\left(\frac{1}{a^2} + \frac{1}{b^2}\right)} \tag{7}$$

where $\delta_{max}$ (m) is the maximum compression of the normal deformation under consideration of elastic–plastic deformation.

By solving the integral Equation (7), one can determine the $\delta_{max}$. Considering the strength characteristics and the impact resistance of various materials in the test, an empirical deformation magnification factor $\kappa$ was given. Substituting $\delta_{max}^* = \kappa\delta_{max}$ into Equation (6), the maximum impact force of the collision after considering the elastic–plastic theory can obtained. After considering the coupling effect of the experimental composite materials, the modified parameter reduction coefficients of LSKLCW were given. The maximum impact force calculation model was obtained through the joint solution process, as shown in Equation (8).

$$F_{max} = \gamma\left[P_y + 2\pi Rp_y\left(\delta_{max} - \delta_y\right)\right] \tag{8}$$

where $F_{max}$ (kN) is the maximum impact force of the collision after considering the elastic–plastic theory and the reduction factor of modified parameter; $\gamma$ is the reduction factor of the modified parameter.

Fitting analysis was based on the mathematical calculation model and test results. We took the W4 specimen as a typical calculation example for fitting, and the results of the fitting are shown below in Table 4.

**Table 4.** Data results comparison of the maximum impact force between test and theoretical calculation.

| Impact Height, m | Test Result, kN | Result of Theoretical Calculation, kN | Contrast Ratio of Theoretical Calculation to Test |
|---|---|---|---|
| 1 | 6.87 | 7.31 | 1.06 |
| 1.5 | 8.79 | 8.98 | 1.02 |

It can be seen from Table 4 that the impact force test value was slightly smaller than the calculated model algorithm value. The analysis shows that there were multiple friction losses in the test, and the elastic modulus of the composite material was different in a complex coupling state. Based on the influence of the above multiple factor, the test result was smaller than the result of the theoretical calculation model. The ratio of the fitting result of the composite wall specimen under the impact height of 1 m was 1.06, and the ratio of the fitting result under the impact height of 1.5 m was 1.02. After verification, the test calculation model was scientific and reasonable and applicable within the allowable error range of the test.

## 6. Conclusions

The analysis of the strain and the displacement time-history curve show that the impact process of the test of the composite wall was roughly divided into three stages: elasticity, elastic–plastic and failure stages. Among them, the elasticity stage and the elastic–

plastic stage occupied most of the duration of the low-speed impact test, and the failure stage was relatively short. The vertical displacement of the most unfavorable collision point of the composite wall and the amount of strain were increased after the velocity of the impact object increased and have a certain positive correlation. Moreover, the law of the elastic–plastic change of the composite wall under the low-velocity impact was fed back through the test, and the analysis basis of the composite wall in the impact resistance research is provided by these results.

The impact resistance of composite wall specimens would be affected by the light steel mesh, light steel framework, concrete strength and other factors. Among them, the influence of concrete strength was more significant. The LSKLCW was still in the elastic working stage under the impact of a 30 kg standard sandbag. The results show that this type structure of composite wall would not produce obvious cracks when under the accidental impact of a weight within 30 kg. The composite wall of this type of combination has good impact resistance. Therefore, in the design of composite walls with low-velocity impact resistance, the strength of the concrete can be mainly improved, which can better improve the impact resistance of the composite wall.

By comparing and fitting the numerical calculation model with the results of the test, the effectiveness of the model and the accuracy of the algorithm were verified by these results. The results of the test could feedback to the actual situation realistically within the range of error, and the numerical model is suitable for the calculation of wall specimens under a low-velocity impact. In addition, the model can calculate the impact force under the low-velocity impact state through the damage state of the composite wall, and can provide an analytical basis and help in the impact detection and impact resistance design of the composite wall. Above all, the results of this paper provide a certain experimental basis and theoretical reference basis for the impact research of composite walls.

Furthermore, this paper studies the impact resistance of LSKLCW under low-velocity impact. However, more trials and analytical research are needed to study the impact resistance of composite walls under medium-velocity and high-velocity impact in the future.

**Author Contributions:** Conceptualization; Methodology; Resources; Software; Validation; Writing—original draft; Writing—review and editing, J.Y., J.Z. and W.Y. All authors have read and agreed to the published version of the manuscript.

**Funding:** This research was funded by the Open Fund of National Joint Engineering Research Laboratory (No. 16BCX01) and the National and Local Joint Engineering Laboratory for Long-Term Performance Improvement Technology of Bridges in Southern Region of Changsha University of Science and Technology of Hunan Province, No. 960.

**Institutional Review Board Statement:** Not applicable.

**Informed Consent Statement:** Not applicable.

**Data Availability Statement:** The data presented in this study are available on request from the corresponding author.

**Acknowledgments:** This research was supported by the Open Fund of National Joint Engineering Research Laboratory (No. 16BCX01) and the National and Local Joint Engineering Laboratory for Long-Term Performance Improvement Technology of Bridges in Southern Region of Changsha University of Science and Technology of Hunan Province, No. 960.

**Conflicts of Interest:** The authors declare no conflict of interest. The funder had no role in the design of the study; in the collection, analyses, or interpretation of data; in the writing of the manuscript, or in the decision to publish the results.

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
