# Peer review of "Analytical Research on the Impact Test of Light Steel Keel and Lightweight Concrete of Composite Wall"

_applsci, doi:10.3390/app12062957_

Round 1

Reviewer 1 Report

Authors have presented their work on "Analytical research on impact test of light steel keel and lightweight concrete of composite wall". Presentation of the article is well written. Experimental results are verified with theoretical model, which makes the study verified. 

Author Response

Response to Reviewer 1 Comments

Dear professor,

Nice to receive your report, we are grateful for your comments and suggestions. And our response as follows:

Point 1: Authors have presented their work on "Analytical research on impact test of light steel keel and lightweight concrete of composite wall". Presentation of the article is well written. Experimental results are verified with theoretical model, which makes the study verified..

Response 1: We are really appreciate your comments and suggestions, and a detailed description was added in the chapter-introduction.

Reviewer 2 Report

You do a good job describing the experiment and the measurements but there is no discussion about the application. After reading the paper I am still left with the question what do the results mean to wall panels? The conclusions do not discuss serial 1, 2, 3, and 4 in sufficient detail to make the results applicable to other situation. Please expand on the conclusions

Author Response

Response to Reviewer 2 Comments

Dear professor,

Nice to receive your report, I am grateful for your comments and suggestions. And each items are responded and explained as follows:

Point 1: You do a good job describing the experiment and the measurements but there is no discussion about the application. After reading the paper I am still left with the question what do the results mean to wall panels? The conclusions do not discuss serial 1, 2, 3, and 4 in sufficient detail to make the results applicable to other situation. Please expand on the conclusions.

Response 1: Some discussions about these conclusions were expanded in the Conclusion to make the results applicable to other situation. And some future directions of this research were described and added in the Conclusion.

Reviewer 3 Report

Interesting written scientific paper. The authors have dealt in this study with the impact test of light steel keel and lightweight concrete of composite wall.

Some minor adjustments are needed for improving the manuscript:

  • The originality and novelty of the research are not good highlighted in the Abstract.
  • Page 1, Please, describe more in the deep state of the art in the Introduction chapter, or add a new section, „Literature review“.
  • Page 2, Chapter-Test overview, The reason of selection this realized test is not highlighted
  • Page 4, Figures 5-7-are poor commented
  • Page 7, Figure 12- replace the text (the description of Figure 12) before Figure 12. It is better for the understanding of readers.
  • Page 12, Conclusion-describe the future direction of your research. It is always important to know how to help this research your next.

Author Response

Response to Reviewer 3 Comments

Dear professor,

Nice to receive your report, I am grateful for your comments and suggestions. And each items are responded and explained as follows:

Point 1: The originality and novelty of the research are not good highlighted in the Abstract.

Response 1: The originality and novelty of the research was appropriately described and added in the abstract. Through the description “four composite wall specimens were designed to conduct dynamic simulation impact tests” show the originality, and the description “the failure mode, time-history curves of strain and displacement were analyzed and studied by test equipment and loading system” show the novelty.

Point 2: Please, describe more in the deep state of the art in the Introduction chapter, or add a new section, “Literature review”.

Response 2: Page 1, A detailed description was added in the chapter-introduction.

Point 3: Page 2, Chapter-Test overview, The reason of selection this realized test is not highlighted.

Response 3: A new reason of selection this realized test was added in the chapter-Test overview. Besides, this test system can timely feedback the real-time changes of the measurement point strain in the test, and these instruments were used frequently in the test because of their intelligent and convenient mechanism.

Point 4: Page 4, Figures 5-7 are poor commented.

Response 4: New description has been reworked in Figure 5-7 and new statements were added.

Point 5: Page 7, Figure 12- replace the text (the description of Figure 12) before Figure 12. It is better for the understanding of readers.

Response 5: The text of the description of Figure 12 was replaced at line 200-222 (Page7) and more reasonable description was given.

Point 6: Page 12, Conclusion-describe the future direction of your research. It is always important to know how to help this research your next.

Response 6: Some future directions of this research were described and added in the Conclusion.

This manuscript is a resubmission of an earlier submission. The following is a list of the peer review reports and author responses from that submission.

Round 1

Reviewer 1 Report

Authors present analytical research on the impact test of light steel keel and lightweight concrete of composite wall. Detailed comments are below:

  1. It needs to provide more information about the experimental setting. For example, on line 76, the specifications of the ‘DH3822’ and ‘LVDT’ measuring equipment are missing. Also, please provide the data measurement interval.
  2. Please provide more information about the material, such as type, thickness, and yield strength of the steel. On Line 308, an ideal elastoplastic condition was assumed for the material. Do you mean perfect plastic material without hardening? Generally, it is not called elatostoplastic material. In this case, authors should clearly specify it is elastic-rigid plastic. Moreover, it is not reasonable because there have been already a lot of models considering material hardening in the dynamic loading conditions. Or authors should provide enough reasons not considering hardening.
  3. Moreover, the terms elasto-plastic, elastic-plastic, and elastoplastic are mixed throughout the manuscript.
  4. Figure 10 is the result of measurement with 'DH3822'. Even though there are just two data points, there is a curved line. How did you get the curved line with just two points.
  5. In Figure 10, only the results of W1 are shown. By adding W2, W3, and W4 data, it is recommended to analyze the results by comparing the strain results up to the position (1-8).
  6. All strain gages are mounted on the area in contact with the sandbag, where seems that compression occurs due to bending during impact. However, in Figure 10, measuring points 2 and 4 represent the tensile strain. Please clarify this issue and discuss more.
  7. In Equation 1, Pe was expressed as elastic contact pressure, and the unit was presented in (kPa). However, it seems that the term on the right side is a formula for calculating the applied force, and is calculated as (N) or (kN) as a unit of force. I am confusing, please clarify this issue.
  8. In Figure 12, the vertical displacement of measuring points 1-5 was measured. If the center of gravity of the sandbag coincides with the center of the specimen, It seems that the maximum value is measured at point 1, and the displacement values ​​at points 2-5 should be the same. What do you think about it?

Reviewer 2 Report

Authors have presented analytical research on impact test of LSKLCW. Presentation of the work is good and there are some areas where it could be improved. 

  1. Introduction can be improved by adding more research background. 
  2. Please re-write section 2.2 and explain in more detail about the test equipment and load system.
  3. In Fig. 11, please zoom the area of the plot to visualize the curves.